# New Insight into the Concanavalin A-Induced Apoptosis in Hepatocyte of an Animal Model: Possible Involvement of Caspase-Independent Pathway

**DOI:** 10.3390/molecules28031312

**Published:** 2023-01-30

**Authors:** Xiangli Zhao, Cheng Fu, Lingjuan Sun, Hao Feng, Peiling Xie, Meng Wu, Xiaosheng Tan, Gang Chen

**Affiliations:** 1Institute of Organ Transplantation, Tongji Hospital, Tongji Medical College, Huazhong University of Science and Technology, Wuhan 430030, China; 2Key Laboratory of Organ Transplantation, Ministry of Education, NHC Key Laboratory of Organ Transplantation, Key Laboratory of Organ Transplantation, Chinese Academy of Medical Sciences, Wuhan 430030, China

**Keywords:** concanavalin A, hepatocyte, cell death, apoptosis, liver injury

## Abstract

Concanavalin A (Con A) is known to be a T-cell mitogen and has been shown to induce hepatitis in mice through the triggering of conventional T cells and NKT cells. However, it remains unknown whether Con A itself can directly induce rapid hepatocyte death in the absence of a functional immune system. Here, by using an immunodeficient mouse model, we found Con A rapidly induced liver injury in vivo despite a lack of immunocyte involvement. We further observed in vitro that hepatocytes underwent a dose-dependent but caspase-independent apoptosis in response to Con A stimulation in vitro. Moreover, transcriptome RNA-sequencing analysis revealed that apoptosis pathways were activated in both our in vivo and in vitro models. We conclude that Con A can directly induce rapid but non-classical apoptosis in hepatocytes without the participation of immunocytes. These findings provide new insights into the mechanism of Con A-induced hepatitis.

## 1. Introduction

Intravenous injection of the plant lectin concanavalin A (Con A) is a widely used model for acute immune-mediated hepatitis in mice [1]. In terms of its underlying mechanism, Con A-induced hepatic injury is primarily driven by the activation and recruitment of T cells to the liver [2]. Although the Con A-induced liver injury mouse model does not completely mimic the pathogenesis and pathological process of autoimmune hepatitis in human patients [3], it is nonetheless a well-established and easy-to-replicate animal model focused on T cell-mediated hepatic damage [4]. T cells, including conventional T cells and natural killer T cells, have been reported to act as triggers and key pathogenic factors in this model [5,6,7,8]. In addition, neutrophils and macrophages are helper cells that promote T cell-induced liver injury [7,9,10]. Thus, this model still has great significance for the development of new therapeutic drugs or other treatment measures to target the immune system, especially T cells.

Unfortunately, as a lectin, Con A not only acts as an activator of the immune system, but it also demonstrates potent activity as a toxin. Since an anti-neoplastic role was reported for Con A, studies have been focused on its autophagic cytotoxicity against tumor cells [6,11,12]. Con A has been shown to bind to surface glycoproteins and glycolipids of many cell types, such as leukocytes [8,13,14], hepatocytes [15] and a large number of transformed and non-transformed cell lines [10,16,17], and it accumulates rapidly in the liver [12,18]. Therefore, it is rational to speculate that Con A also has direct toxicity against parenchymal cells, especially hepatocytes. To date, the toxicity of Con A against hepatocytes has received little attention. Previous studies have revealed that Con A shows cytotoxicity against murine hepatocytes when tested at a high treatment dose (≥50 μg/mL in vitro and 40 mg/kg in vivo) and has a long processing time (≥8 h) in vitro and in vivo [11,19,20]. In contrast, short-term toxicity of Con A at conventional doses (15 to 25 mg/kg) [1] has not been observed.

The established view of the direct toxicity of Con A for hepatocytes has been somewhat obscured by the complexity of the Con A-induced hepatitis model in mice. In addition to activating the immune system, Con A can, in principle, induce blood clotting and aggregation of red blood cell akin to that seen with other lectins such as Phytohemagglutinin (PHA) [14,21]. Nevertheless, depletion of platelets is not protective against Con A-induced liver disease, supporting arguments against passive organ damage as a result of impaired circulation or hypoxia [13,22]. Previous studies have shown that hepatic DNA fragmentation is an early event in Con A-injected mice [15,23], indicating the rapid cytotoxicity of Con A to hepatocytes. Hepatocyte death induced by Con A only at high concentrations or after prolonged treatment does not explain these early events [16,20], and, the time course and mechanism of Con A-induced death of hepatocytes, especially in the early phase, remain unclear.

Our aim in the present research was to elucidate the biological activity and mechanism(s) of Con A hepatotoxicity in the absence of a functional immune system. Using immunodeficient mice, we investigated the hepatotoxicity of Con A in vivo at a conventional lethal dose as reported in the literature, and monitored the progression of Con A-induced hepatocyte death and explored its mechanistic pathway in vitro. Furthermore, we performed RNA sequencing in liver tissues from in vivo experiments and hepatocytes from in vitro experiments, analyzing the changes in the transcriptome before and after Con A exposure and identifying key transcription factors involved. Our findings shed new light on the mechanistic basis of Con A-induced hepatitis.

## 2. Results

### 2.1. Con A Induces Liver Injury in NOD SCID Mice

To exclude the effects of immune system activation, we used NOD SCID mice to detect Con A hepatotoxicity in vivo. T and B cells are absent from NOD SCID mice, and these mice also show functional defect in their macrophages and NK cells [24]. Therefore, using this approach, the main population of Con A-activating immunocytes normally present in wild-type mice was eliminated. First, we injected NOD SCID mice with Con A at various doses and observed the resulting levels of ALT and AST. We characterized the doses according to the terms commonly used for doses administered to wildtype mice: sublethal dose (20 mg/kg), lethal dose (25 mg/kg), and a positive control dose (40 mg/kg) [1,19].

As in a previous study [19], the levels of alanine transaminase (ALT) and aspartate aminotransferase (AST) were dramatically elevated at 8 h after Con A injection in the positive control group (40 mg/kg) (Figure 1A). The lethal dose group showed a moderately severe liver injury in response to Con A injection (Figure 1A). No fluctuation in ALT and a slight increase in AST were observed in the mice injected with Con A at 20 mg/kg (Figure 1A). To investigate further the dynamic changes seen in liver injury in response to Con A, we injected NOD SCID mice with 25 mg/kg Con A and then measured serum ALT/AST levels at various time points (0, 1, 2, 4, 8, and 12 h). The levels of ALT/AST at 1 to 4 h were higher than those that at 0 h, and they were further markedly elevated at 8 to 12 h (Figure 1B). The serum levels of lactate dehydrogenase (LDH) also showed a consistent trend in Con A-treated mice (Figure 1B). As neutrophils have been reported to play an important role in Con A-induced hepatic damage [9], we depleted the neutrophils of the NOD SCID mice by intraperitoneally administering RB6-8C5 24 h before the Con A injection. Surprisingly, depleting the NOD SCID mice of neutrophils did not protect the mice from Con A-induced liver injury (Figure 1C). When wildtype BALB/c mice were injected with Con A (at 25 mg/mg), their serum ALT/AST levels were slightly elevated at 1 to 4 h similar to those of the NOD SCID mice, and there was a sharp increase at 8 h and 12 h (Figure 1D). At 12 h, the average ALT level of the wild-type mice was nearly four times that in the immunodeficient mice, and the average AST level was more than twice that of the immunodeficient mice, indicating that immune system activation started to play an important role in the liver injury of wild-type mice at a later time.

### 2.2. Con A-Injected NOD SCID Mice Exhibit Histological Changes Typical of Liver Injury

To further confirm the hepatotoxicity of Con A in the absence of immunocytes, we evaluated hematoxylin-eosin (H&E)-stained liver tissue sections for histopathology before and after Con A injection. As shown in Figure 2A, massive congestion coincident with cellular swelling was seen by light microscopy at 1 h after Con A injection. The hyperemia in the hepatic sinusoids and hepatocyte edema remained aggravated at 2 and 4 h after Con A administration (Figure 2A). At 8 h after Con A injection, several instances were observed of bridging coagulative death of hepatocytes (Figure 2A).

We then used TdT-mediated dUTP nick-end labeling (TUNEL) staining to analyze hepatocyte apoptosis in the livers of the mice. The area of apoptotic cells was significantly enlarged at 8 h after Con A treatment (Figure 2B). However, the few apoptotic cells observed at 1 h after Con A injection might be attributed to the minor degree of liver injury occurring in the early period after. We also detected infiltration of CD68-positive macrophages and myeloperoxidase (MPO)-positive neutrophils at various times after injection. There was no difference in the numbers of macrophages or neutrophils between the two groups over the course of the experiment (Figure 2B,C). In addition to hepatocyte apoptosis, the classical damage-associated molecular pattern (DAMP) high-mobility group box 1 (HMGB1) molecule was found to be released by the hepatocytes after Con A treatment in the cases of both the immunocompetent and the immunodeficient mice (Figure 2D). Taken together, these pieces of evidence suggest that the injection of Con A can induce rapid liver injury in immunodeficient mice that cannot be attributed to the immune system.

### 2.3. Con A Treatment Leads to Hepatocyte Apoptosis In Vitro

Next, we asked whether Con A could induce hepatocyte apoptosis in vitro; such apoptosis was documented via the externalization of membrane phosphatidylserine, as measured by the binding of annexin V in flow cytometry (FCM). We used alpha mouse liver 12 (AML12) hepatocytes for these in vitro experiments. This cell line is derived from normal murine hepatocytes and is suitable for application in toxicology research. As shown in Figure 3, a rapid apoptosis of hepatocytes was observed in response to Con A treatment (≥20 μg/mL). The Con A-induced apoptosis was found to be dose-dependent, with the high-dose treatment group (40 μg/mL) exhibiting a higher percentage of cells that were positively stained for annexin V compared with the group treated with Con A at 20 μg/mL (Figure 3B). However, no time dependence was observed in this in vitro model. In view of the pivotal role of caspase activation in cell death, we also tried to determine whether caspase was involved in this model. For this purpose, we used various caspase inhibitors targeting pan-caspase (Z-VAD(OMe)-FMK), caspase-3 (Z-DEVD-FMK), and caspase-1 (Ac-YVAD-cmk) to inhibit Con A-induced hepatocyte death. Unfortunately, none of them protected the hepatocytes from Con A-induced death. This evidence indicates that Con A induces dose-dependent hepatocyte apoptosis that is not associated with caspase activation.

### 2.4. The Expression of Apoptosis-Related Genes Is Elevated in Response to Con A Treatment Both In Vivo and In Vitro

After confirming that the hepatocytes were highly sensitive to Con A stimulation, we were interested in exploring the expression of apoptosis-related genes in our in vivo and in vitro models. To this end, we collected liver tissues from NOD SCID mice before and after Con A administration, and AML12 cell pellets before and after Con A stimulation, for further transcriptome RNA-sequencing (RNA-Seq) analysis or characterization. According to the values for fragments per kilobase of exon per million mapped reads (FRKM), at a -fold change >1 and false discovery rate (FDR) of <0.05, 1562 transcripts were upregulated in Con A-injected liver and Con 123 A-treated hepatocytes, when compared with the corresponding control groups (Figure 4A,B). We then analyzed the commonly upregulated genes with the use of gene sets enrichment analysis (GSEA) and gene ontology (GO) enrichment analysis. Upregulated genes from both the in vivo and in vitro models were enriched in the apoptosis-related category according to assessment of the hallmark gene sets by GSEA (Figure 4C). In the GO analysis, we identified a series of terms related to cell death and the apoptotic pathway (Figure 4D). In addition, in the analysis of the lipid profile in the liver, we found that the down-regulated genes in Con A-treated NOD SCID mice were enriched in a series of items related to lipid metabolism (Appendix A). This result suggests the aberrant lipid accumulation in Con A-treated mice, which may lead to liver injury.

To study the mechanism and the connection between the in vivo and in vitro models, we further selected the genes that were upregulated in both models after Con A treatment. Fifteen genes were found to be elevated in Con A-injected NOD SCID mouse livers and also in Con A-treated AML12 cells, according to transcriptome RNA-Seq analysis (Figure 5A). We verified the expression of these genes by real-time quantitative PCR and found that three genes, Zc3h12a, Atf-3, and Csrnp-1, were significantly upregulated after Con A treatment (Figure 5B). To analyze the connection between these three genes and hepatocyte apoptosis, we constructed a protein-protein interaction (PPI) network to identify and locate tightly connected modules. As a result, we discovered that Zc3h12a and Atf-3 are related to a series of apoptotic genes (Figure 5C). Thus, these two genes may be involved in Con A-induced hepatocytotoxicity in mice.

## 3. Discussion

Con A is known to be a T-cell mitogen and has been shown to induce hepatitis in immunocompetent mice through the triggering of conventional T cells and NKT cells [25,26]. However, little attention has been paid to its acute toxicity in parenchymal hepatic cells. In the present study, we demonstrated the rapid cytotoxicity of Con A to hepatocytes both in vivo and in vitro. Liver injury was observed as early as 1 h after Con A administration in immunodeficient mice, at a dose of Con A lethal when injected into wild-type mice, and in vitro experiments, revealed that Con A treatment directly and rapidly triggered hepatocyte apoptosis in a dose-dependent manner in the range of 20 and 40 μg/mL.

It has been reported that many factors contribute to Con A-induced liver injury, particularly the immune system [2]. In addition to the central role of T/NKT cells in Con A-induced liver injury, other leukocyte subsets have also been found to be involved in the process of liver damage, e.g., macrophages [10] and neutrophils [9]. NK cells, but not B cells, have also been reported to accumulate and become activated in Con A-treated mice, but their roles in this model have not been fully elucidated [27]. Here, by using immunodeficient mice lacking T and B cells and with deficiencies in macrophage and NK cell function, we were able to exclude major immunocyte-induced damage. Given that recruitment of neutrophils has been reported to play an essential role in Con A-induced hepatitis in immunocompetent mice, it is noteworthy that further depletion of neutrophils in the NOD SCID mice did not alleviate the liver damage that resulted from Con A administration. Our use of immunodeficient mice and antibody depletion here made it possible to actually observe visually the extent of Con A toxicity to hepatic parenchymal cells.

To date, the toxicity of Con A has received relatively little attention. Con A-induced cytotoxicity towards a variety of cell lines was reported 40 years ago [17], but has not since been characterized in greater detail. Only occasionally have studies pointed to the toxicity of Con A for hepatocytes, and the time course and mechanism of the process have not yet been revealed. A previous report showed that Con A at relatively high concentrations (>= 50 μg/mL) or after a long period of exposure (more than 12 h) can induce hepatocyte death in vitro [20]. Experiments in vivo have revealed that Con A administration leads to elevation of ALT and AST at high doses (40 mg/kg) in NOD SCID mice [19]. However, the treatment time and dosage used in that previous research did not correspond closely to the conditions in most studies of Con A, in which it was applied as an immunostimulant.

Our current findings reveal that Con A-induced apoptosis of hepatocytes in vitro can be observed as early as 1 h after Con A administration, and the dose (>= 20μg/mL) for the induction of hepatocyte apoptosis is lower than that reported in the previously mentioned study [19]. The differences between our findings and those of previous work may be attributed to the choice of method for detecting hepatocyte apoptosis. Flow cytometry with annexin V staining and immunofluorescence using TUNEL staining are likely to be more sensitive in detecting apoptotic cells than lactate dehydrogenase (LDH) release assays and MTT assays. In the present study, the comparable ALT and AST levels between NOD SCID and BALB/c mice at early stages (<= 4 h after Con A injection) suggest that Con A cytotoxicity is the major cause of early liver injury. In contrast, fulminant liver injury that was observed to begin 8 h after Con A injection in the immunocompetent mice implies that activation of the immune system was involved in the advanced stages of the model.

Caspase-dependent apoptosis has been reported to be a key factor in hepatocyte cell death, especially in the Con A-induced acute liver-failure model [28]. Here, we report an enrichment in multiple apoptotic related pathways among the upregulated genes in the Con A-treated group compared with the control group for both our in vitro and in vivo experiments, indicating that apoptosis is involved in Con A-induced hepatocyte death. Furthermore, our in vitro experiments indicate that Con A-induced hepatocyte apoptosis is caspase-independent, suggesting that a non-classic apoptotic process is involved in this model. By comparing the upregulated genes from the in vitro and in vitro experiments, we identified three genes (Zc3h12a, Atf-3, and Csrnp-1) that were commonly upregulated in response to Con A and may be involved in Con A-induced apoptosis of hepatocytes. These genes have been reported to be related to apoptosis in many cell types and mouse models [29,30,31,32,33,34,35,36,37,38,39,40,41]. To determine the roles played by these three genes in Con A cytotoxicity, further studies should be carried out involving overexpression and silencing of these genes in hepatocytes.

## 4. Materials and Methods

### 4.1. Reagents and Cell Lines

Concanavalin A (C2010) was purchased from Sigma-Aldrich (St. Louis, MO, USA) and dissolved in saline solution at a concentration of 4 μg/μL. RB6-8C5mAb was purchased from BioxCell (Lebanon, NH, USA), and Z-VAD(OMe)-FMK, AC-YVAD-cmk, and Z-DEVD-FMK were purchased from MCE (Monmouth Junction, NJ, USA). The working concentrations of Z-VAD(OMe)-FMK, AC-YVAD-cmk, and Z-DEVD-FMK were 50 μM, 80 μM, and 100 μM, respectively, according to articles previously reports [42,43,44]. Alpha mouse liver 12 (AML12) hepatocytes were obtained from the American Type Culture Collection (ATCC CRL-2254TM). AML12 hepatocytes were maintained at 37 °C in DMEM/F-12 (Gibco) supplemented with 10% fetal bovine serum (Yeasen, Shanghai, China) and penicillin/streptomycin (final concentration, 100 U/mL and 0.1 mg/mL, respectively) in a 5% CO_2_ humidified incubator.

### 4.2. Mice and Treatment

Specific pathogen-free BALB/c (6–8 weeks old, 18–23 g) and NOD.CB17-Prkdcscid/NcrCrl (NOD SCID) mice (6–8 weeks old, 20–28 g) were purchased from Charles River Laboratories (Beijing, China). All mice were maintained under controlled conditions (22 °C, 50% humidity, 12 h light/dark cycle, with lights on at 7:00 AM). BALB/c and NOD SCID mice were injected intravenously with saline solution or Con A. A typical dose-identification experiment sequentially tested the liver-injury-inducing abilities of 20 mg/kg, 25 mg/kg, and 40 mg/kg, each dissolved in saline solution [1,19]. To avoid liver injury caused by intravenous injection, the total volume of Con A did not exceed 150 μL per mouse. Mice were sacrificed at 1, 2, 4, 8, or 12 h after Con A treatment. For depletion of neutrophils, a single dose of RB6-8C5mAb (250 μg) was injected intraperitoneally 24 h before Con A administration [9]. The mouse care and experimental protocols used in this study were approved by the Huazhong University of Science and Technology Animal Care and Use Committee (TJH-202208010, 28 October 2022).

### 4.3. Serum Biochemical

Animals were not fasted before sacrifice. Under mild pentobarbital anesthesia, blood samples were collected from the orbital venous plexus of mice using 1.5 mL enzyme-free centrifuge tubes. The serum was separated by centrifugation at 12,000× *g* for 10 min at room temperature. As a measure of hepatocellular injury, the activities of alanine aminotransferase (ALT), aspartate aminotransferase (AST), and lactate dehydrogenase (LDH) were measured with an automated biochemical analyzer BS-200 (Mindray, Shenzhen, China). The linear ranges for the detection kit of ALT, AST, and LDH were 4-1000 U/L, 4-800 U/L, and 4-1000 U/L, respectively. The samples were diluted at proper rates before detection.

### 4.4. Histological Examination

For histological assessment, liver tissues were fixed in 4% paraformaldehyde for at least 24 h. After fixation, they were embedded in paraffin, and 4-μm thick sections were then stained with hematoxylin and eosin (H&E) for examination of tissue damage under light microscopy.

### 4.5. Immunohistochemistry and Immunofluorescent Staining

Liver sections were stained with anti-CD68 (GB113109, Servicebio, Wuhan, China), anti-HMGB1(GB11103, Servicebio, Wuhan, China), or anti-myeloperoxidase (GB11224, Servicebio) according to a standard protocol. Apoptotic cells were evaluated using a TUNEL assay kit according to the manufacturer’s instructions (Servicebio, Wuhan, China). All the stained sections were observed under light microscopy, and images were collected by fluorescence microscopy with a Nikon Eclipse C1. For relative quantification of immunohistochemistry and immunofluorescent stained sections, three representative images from each section were analyzed.

### 4.6. Flow Cytometry

AML12 cells were treated with various concentrations of Con A as indicated then stained for allophycocyanin-conjugated-Annexin V at 1, 2, or 3 h after Con A treatment, then analyzed with a BD Celesta cell analyzer (BD Biosciences, Franklin Lakes, NJ, USA). Flow cytometry data were analyzed using FlowJo software (TreeStar, Ashland, OR, USA).

### 4.7. RNA-Seq Analysis

Total RNA was extracted from Con A-injected NOD SCID mouse liver or Con A-treated AML12 cells using TRIzol (Invitrogen, Carlsbad, CA, USA). Total RNAs (2 μg) were used for preparation of a stranded RNA sequencing library by means of a stranded mRNA library prep kit from Illumina (DR08502, Seqhealth), following the manufacturer’s instructions. The library products corresponding to 200–500 bp were enriched, quantified, and then sequenced on a Hiseq X 10 sequencer (Illumina). Differences in the gene expression profiles of Con A-injected NOD SCID mouse liver and Con A-treated AML12 cells were determined by RNA-Seq data analysis (Bioyigene, Wuhan, China). In brief, raw sequencing data were first filtered by FastQC; low-quality reads were discarded, and adaptor sequences were trimmed. After quality filtering, each sample had ~51.3–69.4 million clean reads. Clean reads from each sample were mapped to the Mus musculus GRCm38 reference genome using hisat2. Significantly differentially expressed transcripts were screened by applying the criteria FC ≥2 or ≤−2 and *p*-value ≤ 0.05. The RNA-Seq data, entitled “Transcriptome RNA-seq analysis for concanavalin A-injected NOD SCID mice liver or Con A-treated AML12 cells” were deposited in the Sequence Read Archive (SRA) with BioProject number PRJNA845812. Gene ontology (GO) analysis of differentially expressed genes was conducted using the GO-seq R package, with a p-adjust (FDR) < 0.05 to determine statistically significant enrichment.

### 4.8. Quantitative RT-PCR

Total RNA was extracted from liver tissues or AML12 cells after treatment as stipulated using a RNAfast200 kit (Fastagen, Shanghai, China), and cDNA was synthesized using the PrimeScript RT reagent kit (Takara, Tokyo, Japan). The expression levels of genes of interest and of the HPRT control were assessed by PCR with SYBR Green mix (Yeasen). Transcript levels of target genes were calculated as the ratio of target gene expression to HPRT expression. Fold changes in target gene expression were analyzed by StepOne software v2.3 (Applied Biosystems, Waltham, MA, USA) using the delta/delta CT method. The sequences for the primers were Zc3h12a (F: 5′-ACGAAGCCTGTCCAAGAATCC-3′, R: 5′-TAGGGGCCTCTTTAGCCACA-3′), Atf-3 (F: 5′-GAGGATTTTGCTAACCTGACACC-3′, R: 5′-TTGACGGTAACTGACTCCAGC-3′), Csrnp-1 (F: 5′-CCGTCTACTATTTCCCACGGT-3′, R: 5′-AACTCAGCTAAGGAGAAAAGGC-3′), Irf-1 (F: 5′-ATGCCAATCACTCGAATGCG-3′, R: 5′-TTGTATCGGCCTGTGTGAATG-3′), KIF-11 (F: 5′-CATGGACATTTGTGAGTCGATCC-3′, R: 5′-CCTTTGGTAGATCAGGTGCAG-3′), Ddit-4 (F: 5′-CAAGGCAAGAGCTGCCATAG-3′, R: 5′-CCGGTACTTAGCGTCAGGG-3′).

### 4.9. Protein-Protein Interaction (PPI) Network

Through the Search Tool for the Retrieval of Interacting Genes (STRING, https://cn.string-db.org/) online database (accessed on 16 June 2022), protein-protein interaction (PPI) networks were established to investigate the physical or functional interaction between proteins that were affected by apoptosis. Cytoscape software (Version 3.8.0, https://cytoscape.org/) (accessed on 18 June 2022) was utilized to analyze and visualize the PPI network and cross-talk pathways of ATF-3 and Zc3h12a.

### 4.10. Statistical Analysis

Data are represented as means ± standard deviation (SD) and were analyzed with Prism version 7.0 (GraphPad Software, San Diego, CA, USA). The data were analyzed by the use of Student’s t-test to compare two groups. Differences were considered significant when *p* < 0.05. *p* values are denoted in the figures as follows: *, *p* < 0.05; **, *p* < 0.01, and ***, *p* < 0.001.

## 5. Conclusions

Overall, we observed that Con A injection led to liver injury in immunodeficient mice in vivo, and Con A treatment induced hepatocyte apoptosis in vitro. Analysis of transcriptome changes further confirmed that apoptotic-related pathways were activated in both mouse liver and liver cell line in response to Con A treatment. Our findings reveal that Con A is directly and rapidly toxic to hepatocytes without any involvement of immunocytes, providing additional insight into the mechanism of Con A-induced hepatitis. Our findings reveal that Con A is directly and rapidly toxic to hepatocytes without any involvement of immunocytes, providing additional insight into the mechanism of Con A-induced hepatitis.

## Figures and Tables

**Figure 1 molecules-28-01312-f001:**
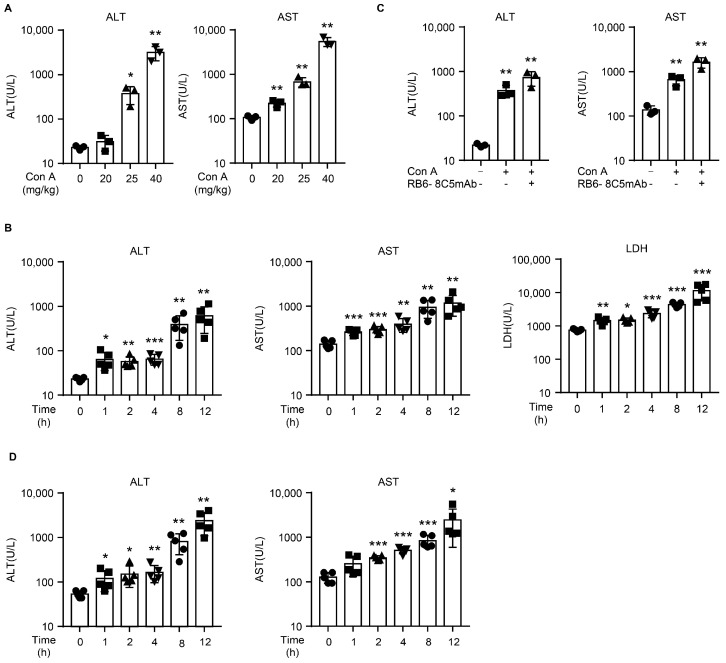
Con A induces liver injury in NOD SCID mice. NOD SCID or BALB/c mice were intravenously injected with Con A (20 mg/kg, 25 mg/kg, or 40 mg/kg of body weight) or saline solution as a mock injection (negative control), and serum samples were collected 1, 2, 4, 8, and 12 h after injection for the following analyses: (**A**) Serum levels of alanine aminotransferase (ALT) and aspartate aminotransferase (AST) 8 h after Con A injection (*n* = 3 for each group). (**B**) The plasma concentrations of ALT, AST, and lactate dehydrogenase (LDH) in NOD SCID mice (*n* = 5 for each group) intravenously injected with Con A at 25 mg/kg of body weight or saline solution at various time points. (**C**) NOD SCID mice (*n* = 3 for each group) pretreated with RB6-8C5 mAb before Con A injection, and the plasma concentrations of AST/ALT detected at 8 h post-Con A treatment. (**D**) Plasma concentrations of ALT/AST in BALB/c mice (*n* = 5 for each group) intravenously injected with Con A at 25 mg/kg of body weight or saline solution at various time points. Bar graphs show means ± SD. Each symbol represents one individual. *, *p* < 0.05; **, *p* < 0.01, and ***, *p* < 0.001 vs. the control group.

**Figure 2 molecules-28-01312-f002:**
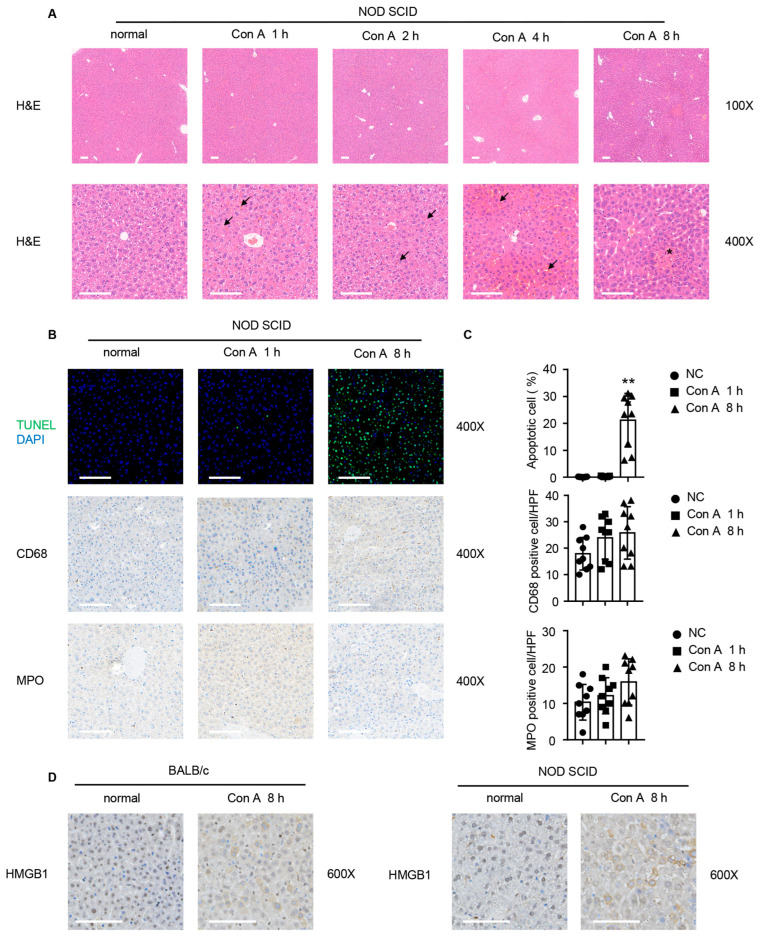
Con A-injected NOD SCID mice exhibit histological changes typical of liver injury. Liver tissues were obtained from NOD SCID or BALB/c mice that had been injected with Con A (25 mg/kg of body weight) or saline solution at the times indicated. (**A**) Histopathological changes in the liver were detected by hematoxylin and eosin (H&E) staining, and representative images are shown (magnification: 100× and 400×). The triangles indicate congestion in the liver tissue, and the asterisk indicate bridging coagulative death of hepatocytes. Scale bar, 100 µm. (**B**) Immunofluorescence was applied to detect the cells that were positively stained by TUNEL in livers from NOD SCID mice, with or without Con A administration at indicated time points (upper panel). Immunohistochemical images of staining for CD68 (middle panel) and MPO staining (lower panel) in liver tissue from NOD SCID mice are shown (magnification: 400×). Scale bar, 100 µm. (**C**) Bar graphs depict the percentage of TUNEL positively staining cells (upper panel), and cell counts for CD68 staining (middle panel), and MPO staining (lower panel) per HPF (*n* = 9 for each group). Each symbol represents one individual. (**D**)Immunohistochemical analysis of HMGB1 in NOD SCID mouse liver sections 8 h after Con A or vehicle administration (magnification: 600×). Scale bar, 100 µm. Bar graphs show means ± SD. **, *p* < 0.01, vs. the normal group.

**Figure 3 molecules-28-01312-f003:**
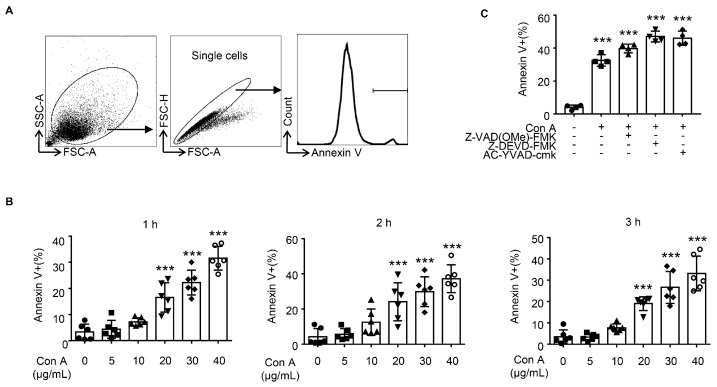
Con A treatment leads to hepatocyte apoptosis in vitro. AML12 cells were treated with or without Con A at various final concentrations for 1, 2, or 3 h, and the cells were then incubated with annexin V. (**A**) Gating strategy for Annexin-positive AML12 cells. (**B**) Bar graphs depict the percentages of AML12 cells positively stained for annexin V after Con A (at various concentrations) or vehicle treatment at 1, 2, or 3 h (*n* = 6 for each group). (**C**) AML12 cells were treated with various caspase inhibitors before Con A (20 mg/mL) treatment and then stained with annexin V. The percentage of apoptotic cells was measured by flow cytometry. The results of quantitative analyses of apoptosis rate are shown. (*n* = 4 for each group). Each symbol represents one individual. Data are shown as means ± SD. ***, *p* < 0.001 vs. the control group.

**Figure 4 molecules-28-01312-f004:**
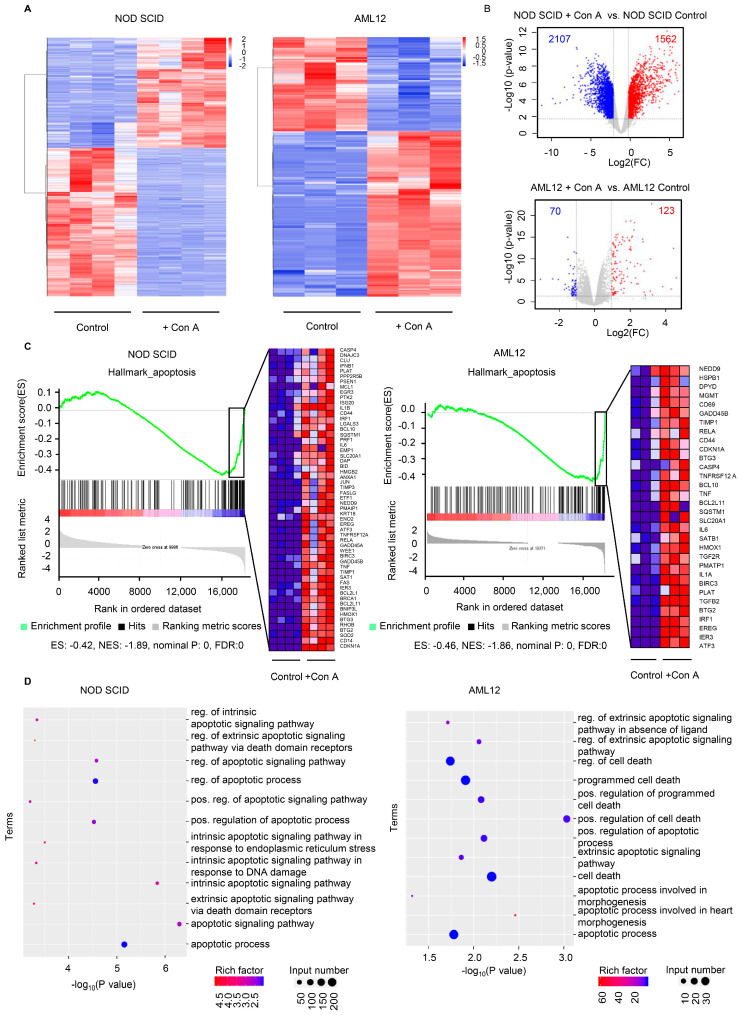
The expression of apoptosis-related genes was elevated in response to Con A treatment both in vivo and in vitro. RNA was obtained from the livers of NOD SCID mice 8 h after Con A (25 mg/kg) or vehicle administration, or from AML12 cells treated for 1 h with Con A (20 mg/mL) or vehicle, then analyzed by RNA-Seq or characterization. (**A**) Heatmap of differentially expressed genes (DEGs). The color code indicates intensity of gene expression from high (red) to low (blue). (**B**) Volcano plot visualizing the DEGs between the control and Con A-treated groups. |Log2FC| ≥ 1 and q < 0.05 were used as the threshold to determine the significance of DEGs. FC, -fold change. Red dots represent genes with increased expression, blue dots represent genes with decreased expression, and gray dots indicate transcripts that did not change significantly. (**C**) Transcripts from the upregulated genes from both the in vivo and in vitro models that were determined to be enriched in apoptosis-related terms according to gene set enrichment analysis (GSEA) analysis. ES, enrichment score; NES, normalized enrichment score; FDR, false-discovery rate. (**D**) Gene ontology (GO) analysis showed the enriched pathways of upregulated genes from the in vivo and in vitro models.

**Figure 5 molecules-28-01312-f005:**
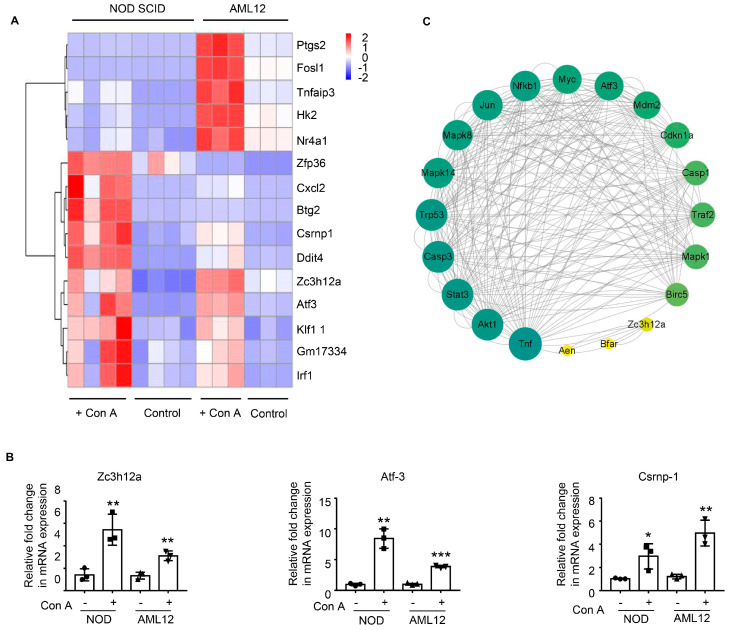
Identification of apoptosis-related genes that were upregulated in response to Con A treatment both in vivo and in vitro. Genes upregulated in Con A-injected NOD SCID liver and Con A-treated AML12 cells were enriched for further analysis. (**A**) Heatmap of elevated genes that were upregulated in both cases. The color code s intensity of gene expression from high (red) to low (blue). (**B**) Quantitative PCR to examine Zc3h12a, Atf3, and Csrnp1 expression after Con A administration or treatment (*n* = 3). (**C**) A PPI network based on the genes involved in apoptosis was constructed to delineate the apoptotic gene interactions. Bar graphs indicate means ± SD. Each symbol represents one individual. *, *p* < 0.05; **, *p* < 0.01, and ***, *p* < 0.001 vs. the control group.

## Data Availability

The RNA-Seq data, entitled “Transcriptome RNA-seq analysis for concanavalin A-injected NOD SCID mice liver or Con A-treated AML12 cells” were deposited in the Sequence Read Archive (SRA) with BioProject number PRJNA845812.

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
