# Peer review of "New Insight into the Concanavalin A-Induced Apoptosis in Hepatocyte of an Animal Model: Possible Involvement of Caspase-Independent Pathway"

_molecules, 2023, doi:10.3390/molecules28031312_

Round 1
Reviewer 1 Report
Dear Authors
Here are minor comments
1. Please, the number of References is too small. Thus, the authors increased the number of references to at least 40~50 by considering the quality of Molecules.
2. Authors revealed that Con-A increased the expression levels of Zc3h12a, Atf-3, and Csrnp-1. Thus, the authors should mention the possible involvement of these genes in the Discussion Part.
3. The contents of 5 Conclusion are too short. and need to be supplemented with more sentences.
4. There are no references in M&M. Please, fill in the references.
5. "4.7 RNA-Seq" is written. Please, add "analysis."
6. Authors just mention "Gibco" in M&M.
Please, add all information (City, State, Country) about each company that provides their project.
Best regards,
Reviewer 2 Report
Chen and coworkers describe in this study the impact of concanavalin A on hepatocyte, and more precisely on apoptosis.
The proposed study is generally well written but, to my opinion, it does not fit with the scopes of “Molecules”. This is strictly biology and nothing in this paper deals with molecular aspects or interactions.
Corrections needed:
- L44 (and others): mL instead of ml.
- L127: treatment.
- L135: the sentence requires not to be hidden by Fig 2.
- L163: involved in this model.
- L204: Three genes were identified as important in the mechanism. Could you precise which proteins they activate? And their functions?
- L235: “In the present study… in vitro”. This sentence is a conclusion and so should appear at the end of the demonstration, not at the beginning.
- L239: dose-dependent manner in the range of XX and XX mg/kg (for instance).
- L250: abbreviate
- L259: In (not in) vivo experiments
- L290-293: this paragraph must be removed.
- L383 (conclusion): the conclusion should be more precise and further developed.
Reviewer 3 Report
In this MS, the authors evaluate the biological activity and mechanism of Concanavalin A (Con A) hepatotoxicity in the absence of a functional immune system in immunodeficient mice. The author found that in vivo, Con A induced rapid liver injury despite a lack of immunocyte involvement. In an in vitro study, the hepatocytes underwent a dose-dependent but caspase-independent apoptosis in response to Con A stimulation. Moreover, transcriptome RNA-sequencing analysis revealed that apoptosis pathways were activated in both in vivo and in vitro models
I think the manuscript is interesting and deserves publication in this journal. There are, however, some comments that I consider major and I'd like to recommend the authors to take in consideration.
- Title: since in this study the authors suggested a mechanism of Con A hepatotoxicity, a further deeper study is required for confirmation. The existing title should be changed to the following for example: ”New insight into the Concanavalin A-induced apoptosis in hepatocyte of an animal model: possible involvement of caspase-independent pathway”.
- Section 4.2: We noticed that the study involved animal models; please add the protocol code and the date of approval.
- Please add a reference for the choice of the Con A doses.
- Please add the number of animals in each treated group.
- Were animals fasted before the sacrifice? From which tube type was blood obtained? Also add the centrifugation speed!
- Certain biochemical marker parameters of hepatotoxicity, in particular LDH, PAL, must be mentioned in this study.
- Determination of ALT and AST”: Please provide their sensitivity and/or detection limit.
- Since the study is focused on the liver, please analyze the lipid profile in the organ.
- All figures should be consistent and professionally drawn. The fonts are different and some of the images are stretched.
- Figure 1: authors should explain the choice of the number of mice, sometimes n=5 and in some other cases n=3, in NOD SCID mice and BALB/c mice?
- Microscopy images: the bar is not clear on the images.
- Figure 2.A (H-E): the authors mentioned (magnification: 100× and 400×). I think it is 200X instead of 400X. Please check. And then, pathological changes in the liver should be clear and reported in Figure 2.A
- Figure 2: How many images were stained and imaged?
- Figure 2. B and C: Immunohistochemistry and immunofluorescent staining: please add histograms showing relative quantification of target proteins.
- Would it be possible to analyze CD45 and/or CD68 in the histological cuts to clearly show the inflammatory cells?
Round 2
Reviewer 3 Report
The authors followed the most of the reviewer's comments and therefore the MS has been improved accordingly.